# Effect of Pregnane X Receptor on *CYP3A29* Expression in Porcine Alveolar Macrophages during *Mycoplasma hyopneumoniae* Infection

**DOI:** 10.3390/ani11020349

**Published:** 2021-01-30

**Authors:** Xiaoyang Yang, Fei Xing, Li Wang, Weimin Zhao, Yanfeng Fu, Feng Tu, Bixia Li, Xiaomin Fang, Shouwen Ren

**Affiliations:** 1Institute of Animal Science, Jiangsu Academy of Agricultural Sciences/Jiangsu Germplasm Resources Protection and Utilization Platform, Nanjing 210014, China; yangxy95v1@163.com (X.Y.); xingjam@163.com (F.X.); wanglids@163.com (L.W.); zhao_weimin1983@aliyun.com (W.Z.); fuyanfeng2011@foxmail.com (Y.F.); tufeng2020@163.com (F.T.); bxlijaas@163.com (B.L.); 2College of Animal Science & Technology, Nanjing Agricultural University, Nanjing 210095, China; 3Institute of Food Safety and Nutrition, Jiangsu Academy of Agricultural Sciences, Nanjing 210014, China

**Keywords:** *Mycoplasma hyopneumoniae*, pregnane X receptor, cytochrome P450 3A29, *IL-6*, *IL-8*

## Abstract

**Simple Summary:**

In the currently intense production process, infection of swine with Mycoplasma pneumonia is common in pig farms around the world, and reduction in the feeding efficiency and the growth rate of sick pigs causes considerable economic losses to the pig-rearing industry. Our study aimed to determine the molecular mechanism by which *Mycoplasma hyopneumoniae* induces inflammation in pigs. Our study showed that *Mycoplasma hyopneumoniae* can regulate the expression of *CYP3A29* by upregulating *PXR* during the inflammatory response induced in porcine alveolar macrophages. These findings may provide useful information for breeding pigs that are resistant to disease.

**Abstract:**

*Mycoplasma hyopneumoniae* (*M. hyopneumoniae*, *Mhp*) is the causative agent of mycoplasma pneumonia of swine (MPS). *M. hyopneumoniae* infection causes inflammation in pigs and leads to considerable economic losses in the pig industry. Pregnane X receptor (*PXR*) is a pluripotent gene regulatory protein that plays an important role in regulating cytochrome P-450 (CYP) in pigs in the context of inflammatory responses, drug metabolism, homeostasis, etc. We previously reported that cytochrome P450 3A29 (*CYP3A29*) expression was significantly upregulated in pigs infected with *M. hyopneumoniae* compared with healthy control pigs. This experiment mainly focused on identifying the role of *PXR* in the regulation of *CYP3A29* and inflammatory factors after *M. hyopneumoniae* infection by establishing pig alveolar macrophage (PAM) cells in which *PXR* was overexpressed or silenced. Our results showed that the overexpression of *PXR* could significantly improve the protein and the mRNA expression levels of *CYP3A29* with and without *M. hyopneumoniae* infection in PAM cells. After the expression of *PXR* was inhibited, protein and mRNA expression levels of *CYP3A29* were significantly reduced with and without *M. hyopneumoniae* infection in PAM cells. Moreover, *PXR* can regulate the mRNA expression levels of *IL-6* and *IL-8* during *M. hyopneumoniae* infection of PAM cells. In conclusion, these results suggest that *PXR* positively regulates *CYP3A29* expression during the inflammatory response caused by *M. hyopneumoniae* infection.

## 1. Introduction

Mycoplasma pneumonia of swine, commonly known as swine asthma, is a chronic respiratory infection in pigs that is caused by *M. hyopneumoniae* and spreads due to close contact with infected individuals [1]. Sick pigs exhibit slow growth and low feed conversion rates and require high use of antibiotics. *M. hyopneumoniae* is an important pathogen that causes great economic losses in the pig industry [2,3]. In recent years, research has focused primarily on the epidemiology and the transmission of this disease and has partially elucidated the interaction between the pathogen and the respiratory tract, and many vaccines have been developed [4]. However, research on the virulence mechanisms and the pathogen–host interactions of *M. hyopneumoniae* is still relatively limited [4]. In particular, the mechanisms by which the host inflammatory response is regulated and immunosuppression is induced have not been fully elucidated. Studying the molecular mechanism by which *M. hyopneumoniae* causes inflammation in pigs is of great significance for the breeding of *M. hyopneumoniae*-resistant pigs.

The porcine *CYP3A29* gene, located on chromosome 3, P16–17, 11 [5], encodes an important metabolic enzyme in the cytochrome P450 superfamily. The mechanism by which *CYP3A29* is regulated in pigs is similar to the mechanism by which *CYP3A4* is regulated in humans [6]. Regarding the relationship between porcine *CYP3A29* and the inflammatory response, studies have shown that the isoenzyme *CYP3A4* participates in the body’s inflammatory response [7,8]. Our previous research used Meishan pigs and Changbai pigs infected with *M. hyopneumoniae* as experimental models, and the results of the expression spectrum microarray screening led to the selection of *CYP3A29* as a candidate gene associated with the response to this infection [9]. It was also found that *PXR* expression increased during *M. hyopneumoniae* infection. Pregnane X receptor (*PXR*), a member of the nuclear receptor (NR) family 1 subfamily, also known as *NR1I2*, is an important regulatory factor in the body and plays an important role in regulating drug metabolism [10,11]. Through ligand binding [12,13], *PXR* plays an important role in the inflammatory response. *PXR* is important for *IFN*-mediated upregulation of *CYP3A29* expression in pig hepatocytes [14,15]. Therefore, we wanted to determine whether *PXR* regulates the expression of *CYP3A29* during *M. hyopneumoniae* infection in pigs.

The purpose of this study was to elucidate the regulatory effect of *PXR* on *CYP3A29* expression during *M. hyopneumoniae* infection. To this end, we prepared porcine alveolar macrophage cells, which play an important role in the pulmonary inflammatory response caused by *M. hyopneumoniae* [16]. Overexpression and silencing of *PXR* in PAM cells before and after *M. hyopneumoniae* infection, which were confirmed by real-time quantitative polymerase chain reaction (RT-qPCR) and western blot analysis, indicated that *PXR* can upregulate the expression of *CYP3A29* during the inflammatory response caused by *M. hyopneumoniae* infection.

## 2. Materials and Methods

### 2.1. Test Materials and Reagents

The porcine alveolar macrophage cell line 3D4/21 (ATCC CRL-2843) [17] and *M. hyopneumoniae* strain JS were kind gifts from the Institute of Veterinary Medicine, Jiangsu Academy of Agricultural Sciences, China; antibiotics (streptomycin and penicillin), fetal bovine serum, trypsin, glutamine, fetal rat serum (FBS), and roswell park memorial institute (RPMI)-1640 medium + 25 mM N-piperazine-N’-[2-ethanesufonic acid] (HEPES) were purchased from Gibco (Grand Island, NY, USA). *CYP3A29* and β-Actin monoclonal antibodies were purchased from Proteintech Group, Inc. (Chicago, IL, USA). Monoclonal antibodies against *PXR* were purchased from Sigma-Aldrich (St. Louis, MO, USA). All the other chemicals were of analytical grade.

### 2.2. Cell Culture

PAM 3D4/21 cell lines were cultured in RPMI-1640 medium containing 10% fetal bovine serum, 100 U/mL penicillin, 100 U/mL streptomycin, 25 mM HEPES, and 1% glutamine at 37 °C with 5% CO_2_.

### 2.3. Construction of the Overexpression Vector and Design of Interference Fragments

*PXR* coding sequence (CDS) were amplified from PAM 3D4/21 cell cDNA and were then ligated into the pMD 18-T vector (Takara, Otsu, Shiga, Japan) using EcoR I and Sal I double-enzyme digestion. The target fragment was inserted into pcDNA3.1 (Invitrogen, Carlsbad, CA, USA) to generate pcDNA3.1-*PXR* using the same double-enzyme digestion. The sequence of the *PXR* siRNA (*PXR*-SUS-992) was synthetized by GenePharma. The sequence was GCAUUAUCAACUUUGCCAATTUUGGCAAAGUUGAUAAUGCTT.

### 2.4. Transfection

Cells transfected with the pcDNA3.1-*PXR* plasmid were used as the experimental group, and cells transfected with the empty pcDNA3.1 vector were used as the negative control group. The cells were transfected with Lipofectamine^TM^ 3000 (Invitrogen, Carlsbad, CA, USA) using 2.5 g plasmid per well for 36 h when the cell density reached 60%. RPMI- 1640 medium containing 10% fetal bovine serum was used in the transfection process. In the interference experiment, the same method was used to transfect the porcine *PXR* interfering fragment siRNA.

### 2.5. M. hyopneumoniae Infection of PAM 3D4/21 Cells

Infection was carried out in a 6-well plate at a density of 2 × 10^5^ cells/well. Two milliliters of 10^6^ CCU/mL *M. hyopneumoniae* strain J was added to each well of the infected group, and 2 mL of RPMI 1640 medium containing 10% fetal bovine serum was added to each well of the control group. The *M. hyopneumoniae* cultures were centrifuged at 12,000× *g* for 20 min, washed with sterile phosphate buffered saline (PBS) twice, and resuspended with antibiotic-free RPMI 1640 medium, and samples were collected 24 h after infection [18]. Infection was confirmed by PCR amplification of the *P36* gene, which encodes a membrane protein of *M. hyopneumoniae*, as previously described [19].

### 2.6. Real-Time Fluorescence Quantitative PCR

Total RNA was extracted from PAM 3D4/21 cells using the HP Total RNA Kit (Omega Bio-Tek, Norcross, GA, USA) according to the instructions. The total RNA concentration was measured by a spectrophotometer. The PrimeScript^TM^ RT reagent Kit (Takara, Otsu, Shiga, Japan) was used for reverse transcription. qPCR was conducted on an ABI 7500 Real-Time PCR System (Applied Biosystems, Foster City, CA, USA) using SYBR Premix Ex Taq^TM^ (Tli RNaseH Plus) (Takara, Otsu, Shiga, Japan). The total amplification reaction system contained 20 μL, and the amplification was carried out at 95 °C for 2 min, followed by 40 cycles of 95 °C for 15 s, 60 °C for 30 s, and 72 °C for 30 s. *HPRT1* was used as the internal reference. Primer sequences are shown in Table 1.

### 2.7. Western Blot

The total protein was extracted by radio immunoprecipitation assay (RIPA) buffer, and the sample concentration was determined by a bicinchoninic acid (BCA) protein assay (Pierce, Thermo Scientific, Waltham, MA, USA). The supernatant was absorbed, mixed with 5× SDS, and denatured at 99 °C for 10 min. SDS-PAGE was performed by adding 30 μg of protein per well to a 10% gel and then transferring the protein to polyvinylidene fluoride (PVDF) membranes. The membranes were blocked with 5% bovine serum albumin (BSA) in 0.1% Tween 20 for 2 h. Then, a rabbit anti-*PXR* antibody (1:500 dilution) and a mouse anti-pig *CYP3A29* antibody (1:1000 dilution) were prepared by diluting the antibodies in 5% BSA; then, the diluted antibodies were incubated with the membranes in the refrigerator at 4 °C overnight. Then, the membranes were washed 3 to 4 times with tris buffered saline tween (TBST), and diluted goat anti-rabbit and goat anti-mouse (1:1000 dilution) antibodies were prepared as secondary antibodies. The membranes were incubated at room temperature for 2 h and washed with TBST. The bands were visualized and quantified with the Clarity Western ECL Substrate (Bio-Rad, Hercules, CA, USA) and quantified by a DNR Bio Imaging System (DNR Industries, Anaheim, CA, USA).

### 2.8. Enzyme-Linked Immunosorbent Assay ELISA

Porcine *IL-6* ELISA Kit (Absin, Shanghai, China) and Porcine *IL-8* ELISA Kit (Jingmei, Yancheng, Jangsu, China) were used to detect the protein expression of *IL-6* and *IL-8* in the cell culture supernatant of the PAM 3D4/21 cells, respectively. 

### 2.9. Statistical Analysis

This experiment used the 2^−△△*Ct*^ method to analyze the data, and *HPRT1* was used as the reference gene. GraphPad Prism was used to generate the graphs and conduct the analyses, and a *t*-test was performed. *p* < 0.05 is indicated by *; *p* < 0.01 is indicated by **; *p* < 0.001 is indicated by ***; and NS indicates no significant difference. Image J was used to detect the gray values of the western blot bands so that the data could be quantitatively analyzed. 

## 3. Results

### 3.1. M. hyopneumoniae Infection Promoted the Expression of IL-6 and IL-8 in PAM 3D4/21 Cells

To evaluate the effect of *M. hyopneumoniae* infection on PAM 3D4/21 cells, the expression of *P36* in PAM 3D4/21 cells was detected (Figure 1A) after stimulation with 100 multiplicity of infection (MOI) *M. hyopneumoniae* for 24 h [20]. Infection was confirmed by PCR amplification of the *P36* gene [19]. P36 is an immunodominant protein encoded by the genome of *M. hyopneumoniae,* and p36 proteins have been shown to be specific for *M. hyopneumoniae* [21]. A single 427-bp band of *P36* was detected by agarose gel electrophoresis, indicating that *M. hyopneumoniae* had successfully infected the PAM 3D4/21 cells. Therefore, infection with 100 MOI *M. hyopneumoniae* for 24 h was used in the subsequent experiments.

We then examined the effect of *M. hyopneumoniae* infection on the expression of *IL-6* and *IL-8*. The results showed that the mRNA expression of *IL-6* (Figure 1B) and *IL-8* (Figure 1C) significantly increased after *M. hyopneumoniae* infection. This result indicates an inflammatory response in *M. hyopneumoniae*-infected PAM 3D4/21 cells.

### 3.2. M. hyopneumoniae Infection Increased the Expression of CYP3A29 and PXR in PAM 3D4/21 Cells

To investigate whether the expression of *CYP3A29* and *PXR* was altered in response to *M. hyopneumoniae* infection, we detected mRNA and protein expression levels of *PXR* and *CYP3A29*. The results showed that the mRNA levels of both *PXR* (Figure 2A) and *CYP3A29* (Figure 2B) significantly increased after *M. hyopneumoniae* infection for 24 h. The protein expression of *PXR* and *CYP3A29* was also upregulated (Figure 2C,D).

### 3.3. PXR Regulates the Expression of CYP3A29 during M. hyopneumoniae Infection of PAM 3D4/21 Cells

#### 3.3.1. Identification and Validation of the pcDNA3.1-PXR Recombinant Plasmid and Efficiency of PXR Overexpression and Interference

The *PXR* recombinant plasmids were identified by restriction analysis and sequencing. Analysis of two fragments released from the recombinant plasmids by digestion with homologous restriction enzymes (EcoR I and Sal I) revealed that the *PXR* gene was correctly inserted. The clone was further confirmed by sequencing.

The efficiency of *PXR* overexpression and silencing was analyzed by assessing the expression of *PXR* mRNA and protein. The *PXR* mRNA and protein levels in PAM 3D4/21 cells were significantly increased after transfection with the pcDNA3.1-*PXR* recombinant plasmid for 36 h (Figure 3A,B). However, the *PXR* mRNA and protein levels in PAM 3D4/21 cells were significantly decreased after transfection with *PXR*-SUS-992 for 36 h (Figure 3C,D).

#### 3.3.2. PXR can Promote CYP3A29 Expression during *M. hyopneumoniae* Infection

To further explore whether *PXR* regulates the expression of *CYP3A29* during *M. hyopneumoniae* infection, mRNA and protein expression of *PXR* and *CYP3A29* were analyzed by qPCR and western blot. The results showed that the overexpression of *PXR* in normal PAM 3D4/21 cells significantly increased mRNA (Figure 4A) and protein (Figure 4B) levels of *CYP3A29*. *PXR* was overexpressed in PAM 3D4/21 cells infected with *M. hyopneumoniae*, and mRNA (Figure 4C) and protein (Figure 4D) levels of *CYP3A29* showed the same trend. When compared Figure 4A,B with Figure 4C,D, it was found that the promotion effect of overexpression of PXR on CYP3A29 with *M. hyopneumoniae* was lower than that without *M. hyopneumoniae* infection. However, silencing of *PXR* in normal PAM 3D4/21 cells significantly decreased mRNA (Figure 5A) and protein (Figure 5B) levels of *CYP3A29*. After transfection of *PXR*-sus-992 into PAM 3D4/21 cells infected with *M. hyopneumoniae*, mRNA (Figure 5C) and protein (Figure 5D) levels of *CYP3A29* were also significantly reduced. Comparing Figure 5A,B with Figure 5C,D, it was found that the inhibiting effect of silencing of PXR on CYP3A29 with *M. hyopneumoniae* was higher than that without *M. hyopneumoniae* infection. It shows that CYP3A29 further increased after *M. hyopneumoniae* infection.

### 3.4. PXR can Promote the Expression of IL-6 and IL-8 during M. hyopneumoniae Infection

Our previous research showed that *M. hyopneumoniae* infection increased the expression of *PXR*, *IL-6,* and *IL-8*. To prove whether *PXR* could affect the expression of *IL-6* and *IL-8* during *M. hyopneumoniae* infection, we detected the expression of *IL-6* and *IL-8* after the overexpression or silencing of *PXR*. The results showed that the overexpression of *PXR* without *M. hyopneumoniae* infection significantly increased the mRNA levels of *IL-6* and *IL-8* (Figure 6A). The silencing of *PXR* without *M. hyopneumoniae* infection decreased the mRNA levels of both *IL-6* and *IL-8* (Figure 6B). Moreover, the overexpression of *PXR* with *M. hyopneumoniae* infection significantly increased the mRNA levels of *IL-6* and *IL-8* (Figure 6C). The silencing of *PXR* with *M. hyopneumoniae* infection decreased the mRNA levels of *IL-6* and *IL-8* (Figure 6D).

Subsequently, we used ELISA to detect the protein expression of *IL-6* and *IL-8* in the cell culture supernatant of the PAM 3D4/21 cells. The results showed that the overexpression of *PXR* without *M. hyopneumoniae* infection increased the protein expression of *IL-8*, but the protein expression of *IL-6* did not change significantly (Figure 7A). The silencing of *PXR* without *M. hyopneumoniae* infection decreased the protein expression of *IL-6* and *IL-8* (Figure 7B). Moreover, the overexpression of *PXR* with *M. hyopneumoniae* infection increased the protein expression of *IL-8*, but the protein expression of *IL-6* did not change significantly (Figure 7C). The silencing of *PXR* with *M. hyopneumoniae* infection decreased the protein expression of *IL-6* and *IL-8* (Figure 7D).

## 4. Discussion

Although mycoplasma pneumonia of swine (MPS) is a serious threat to pig production, the mechanism by which the inflammatory response to *M. hyopneumoniae* is regulated is rarely studied. It was reported that *M. hyopneumoniae* infection could cause inflammation in live pigs [22] and induce the secretion of *IL-6* and *IL-8* from PAM 3D4/21 cells [23], which is consistent with our research. In our study, *CYP3A29* and *PXR* were upregulated after *M. hyopneumoniae* infection in PAM 3D4/21 cells. *CYP3A29*, an important cytochrome P450 (CYP) enzyme, is one of the key candidate genes involved in the regulation of inflammation induced by *M. hyopneumoniae* infection [9]. However, it was not clear how *M. hyopneumoniae* regulates *CYP3A29*-induced inflammation. It is well known that the activation of nuclear receptors can increase the expression and the activity of CYPs [24]. *PXR* is the major transcriptional regulator of cytochrome P450 [13], and it mediates responses to various xenobiotics and endogenous chemicals [25]. Studies have found that *PXR* can positively regulate the interferon-mediated expression of *CYP3A29* in pig hepatocytes [14,15], and cecropin B inhibits the expression of *CYP3A29* by regulating *RXR-α* and *PXR* in pig liver cells [26]. It is speculated that *PXR* may regulate the expression of *CYP3A29* during mycoplasma infection. In our study, the overexpression and the silencing of *PXR* increased and decreased the expression of *CYP3A29* in normal PAM 3D4/21 cells, respectively. In addition, *M. hyopneumoniae*-infected normal PAM 3D4/21 cells significantly increased gene and protein expression levels of *CYP3A29*, while the silencing of *PXR* decreased the expression of *CYP3A29*, and the overexpression of *PXR* increased the expression of *CYP3A29*. Our study is consistent with previous studies, and based on the pathogen infection pattern, our study validated that *PXR* could upregulate the expression of *CYP3A29* during *M. hyopneumoniae* infection in PAM cells.

In our study, we also found that *PXR* can regulate the mRNA expression of *IL-6* and *IL-8*. The silencing of *PXR* in PAM 3D4/21 cells can reduce the mRNA expression of *IL-6* and *IL-8*. When PAM 3D4/21 cells were infected with *M. hyopneumoniae*, the mRNA expression levels of *IL-6* and *IL-8* increased, while the mRNA expression levels of *IL-6* and *IL-8* decreased after *PXR* silencing. In addition, the mRNA expression levels of *IL-6* and *IL-8* increased after the overexpression of *PXR*. Previous studies using atrazine to treat quail hearts found that *CYP3A4* expression induced by the *PXR*/*CAR* pathway and *NF-κB* pathway upregulates *IL-6* and *IL-8* expression [27]. *IL-6* can upregulate the expression of *CYP2C33* through *CAR/RXRα* heterodimers in primary porcine hepatocytes [28]. *IL-6* is an important proinflammatory cytokine that has been shown to affect CYP450 expression in humans [29]. Because there are many members of the CYP450 family, their biological functions may be different. The expression of *CYP3A29* varies among different tissues and ages and is influenced by various factors, such as drugs and inflammatory stimuli [30,31,32]. The role of *PXR* in pigs may be more complex, and *PXR* may react differently to ligands in human and other animal models than it does in pigs [33,34]. For instance, our previous study suggested that *CYP1A1* can inhibit the inflammatory response caused by *M. hyopneumoniae* infection through the *PPAR-γ* signaling pathway in PAM cells [23]. Therefore, our results suggest that, during the inflammatory response to *M. hyopneumoniae* infection in pigs, *PXR* may regulate the expression of the proinflammatory cytokines *IL-6* and *IL-8* by regulating the expression of *CYP3A29*.

In the cell culture supernatant of the PAM 3D4/21 cells, the protein expression of *IL-8* showed the same trend as the mRNA expression. These results indicate that *PXR* does have a regulatory effect on *IL-8* during *M. hyopneumoniae* infection. The protein expression of *IL-6* did not change significantly in the overexpression of *PXR* with and without *M. hyopneumoniae* infection. The protein expression of *IL-6* was down-regulated after *PXR* silencing with and without *M. hyopneumoniae* infection. These results suggested that *PXR* overexpression had a higher regulation effect on *IL-6* at mRNA level than at protein level in PAM cells. Moreover, during the inflammatory response caused by *M. hyopneumoniae* infection, the expression of inflammatory factors may also have other regulatory effects on molecular pathways. Studies have reported that many pathways are involved in regulating the expression of inflammatory cytokines mediated by mycoplasmal lipopeptide Pam2CGDPKHPKSF (FSL-1) in pig monocyte-derived macrophages (MDMs); these pathways include the *TNF* signaling pathway, the cytokine–cytokine receptor interaction pathway, the Toll-like receptor signaling pathway, the janus kinase-signal transducer and activator of transcription (JAK-STAT) signaling pathway, the chemokine signaling pathway, the nucleotide oligomerization domain (NOD)-like receptor signaling pathway, and the *NF-κB* signaling pathway [35]. Furthermore, previous reports have shown that activated *PXR* inhibits the activity of *NF-κB* and the expression of the *NF-κB* target gene *IL-6* [36]. The nuclear factor *NF-κB* family is a key regulator of inflammatory, innate immune, and adaptive immune responses [37,38]. Therefore, the expression of the proinflammatory cytokines *IL-6* and *IL-8* may be regulated by multiple pathways. The relationship between cytochrome P450 enzymes and inflammation should be further studied and will be helpful for the treatment of inflammation. Future studies should investigate the relationship between the *NF-κB* pathway and *PXR* during the inflammatory response to *M. hyopneumoniae* infection. Further elucidation of the molecular mechanism by which mycoplasma causes inflammation in the body is necessary.

## 5. Conclusions

In conclusion, our study found that *PXR* regulates the expression of *CYP3A29* during the course of *M. hyopneumoniae* infection and inflammatory response through inflammatory factors *IL-6* and *IL-8*. This study provides a theoretical molecular basis for the treatment of MPS and for the breeding of pigs resistant to this disease.

## Figures and Tables

**Figure 1 animals-11-00349-f001:**
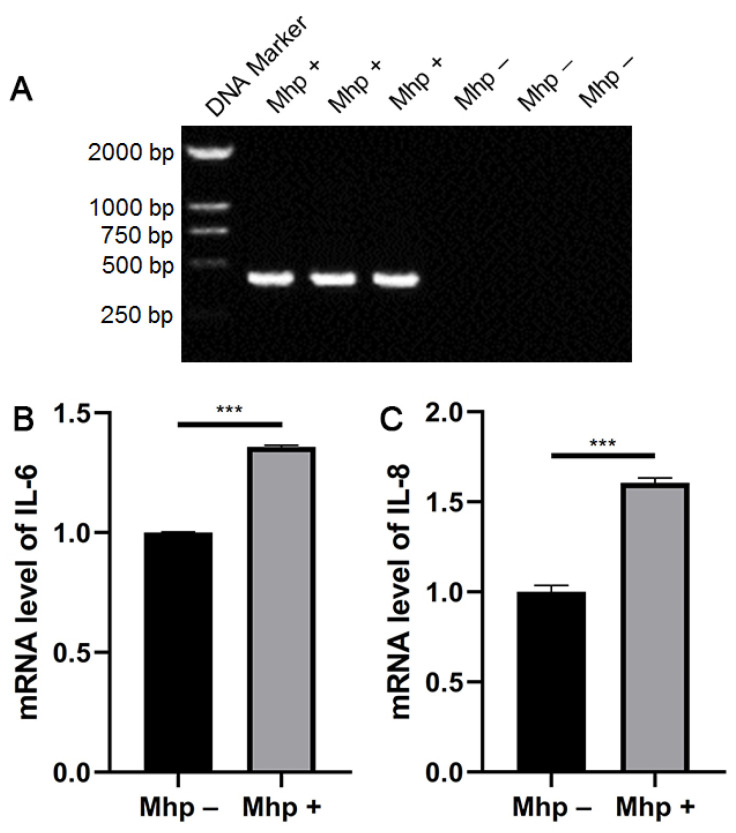
PCR identification of pig alveolar macrophage (PAM) 3D4/21 cells infected with *M. hyopneumoniae* and the mRNA expression of *IL-6* and *IL-8*. (**A**) PCR identification of PAM 3D4/21 cells infected with *M. hyopneumoniae*. (**B**) *IL-6* and (**C**) *IL-8* mRNA expression after *M. hyopneumoniae* infection. The data are presented as the mean ± SD. The experiment was repeated three independent times. *** *p* < 0.001 compared to the uninfected group.

**Figure 2 animals-11-00349-f002:**
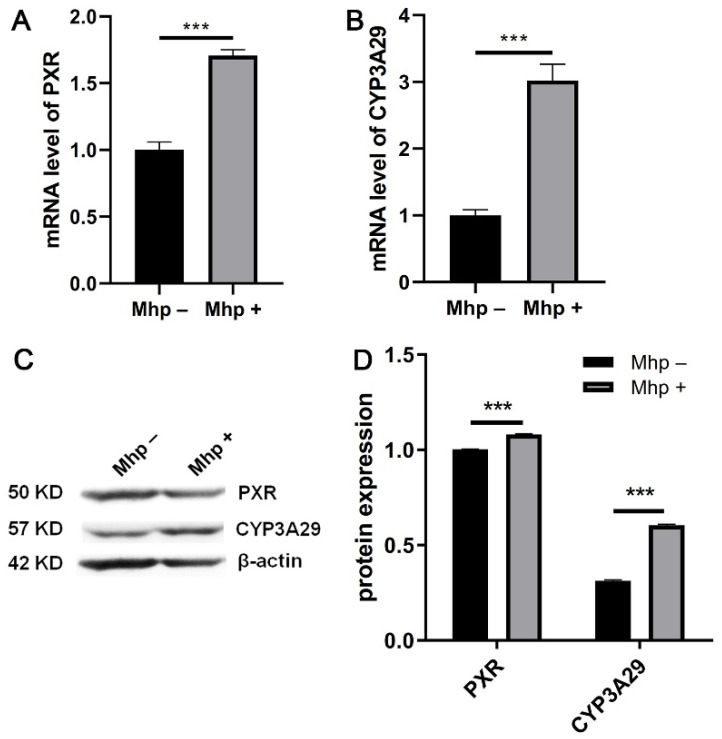
Changes in *PXR* and *CYP3A29* protein and mRNA expression levels after *M. hyopneumoniae* infection. The mRNA expression levels of (**A**) *PXR* and (**B**) *CYP3A29* after *M. hyopneumoniae* infection of PAM 3D4/21 cells for 24 h. (**C**) The protein expression of *PXR* and *CYP3A29* after *M. hyopneumoniae* infection. (**D**) Analysis of the relative protein expression based on the gray values of the bands in panel C. The data are presented as the mean ± SD. The experiment was repeated three independent times. *** *p* < 0.001 compared to the uninfected group.

**Figure 3 animals-11-00349-f003:**
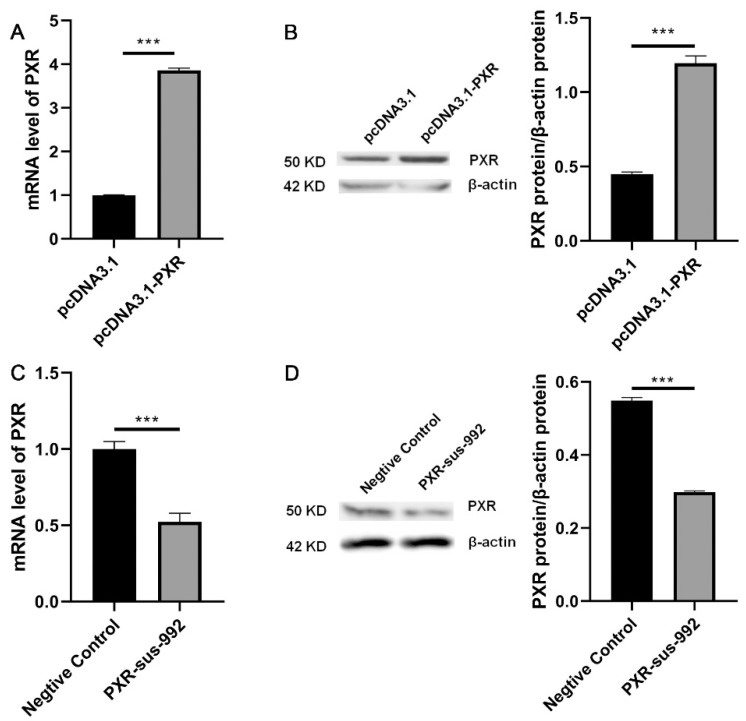
Detection of *PXR* overexpression and interference efficiency. (**A**) Relative mRNA levels of *PXR* after pcDNA3.1-*PXR* transfection into PAM 3D4/21 cells for 36 h. (**B**) pcDNA3.1 or pcDNA3.1-*PXR* was transfected into PAM 3D4/21 cells. The protein expression of *PXR* was measured by western blot. (**C**) Relative mRNA level of *PXR* after *PXR*-sus-992 transfection into PAM 3D4/21 cells for 36 h. (**D**) Negative control or *PXR*-sus-992 was transfected into PAM 3D4/21 cells. The protein expression of *PXR* was measured by western blot. The data are presented as the mean ± SD. The experiment was repeated three independent times. *** *p* < 0.001.

**Figure 4 animals-11-00349-f004:**
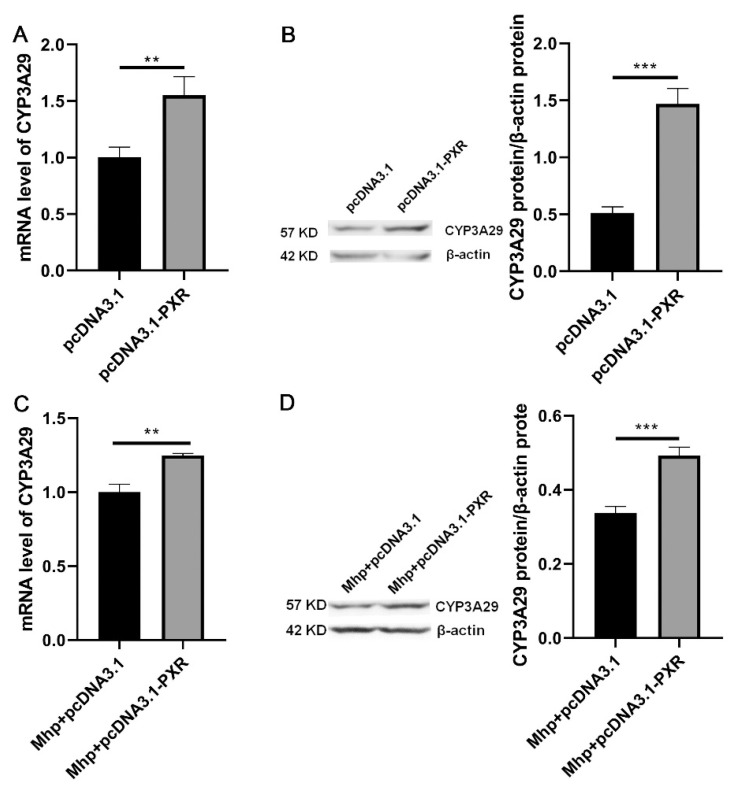
Effect of *PXR* overexpression on *CYP3A29* expression in PAM 3D4/21 cells with or without *M. hyopneumoniae* infection. (**A**) The mRNA expression level of *CYP3A29* after *PXR* overexpression. (**B**) pcDNA3.1 or pcDNA3.1-*PXR* was transfected into PAM 3D4/21 cells. The protein expression of *CYP3A29* was measured by western blot. (**C**) The mRNA expression level of *CYP3A29* after *PXR* overexpression in PAM 3D4/21 cells infected with *M. hyopneumoniae*. (**D**) pcDNA3.1 or pcDNA3.1-*PXR* was transfected into PAM 3D4/21 cells infected with *M. hyopneumoniae*. The protein expression of *CYP3A29* was measured by western blot. The data are presented as the mean ± SD. The experiment was repeated three independent times. ** *p* < 0.01; *** *p* < 0.001.

**Figure 5 animals-11-00349-f005:**
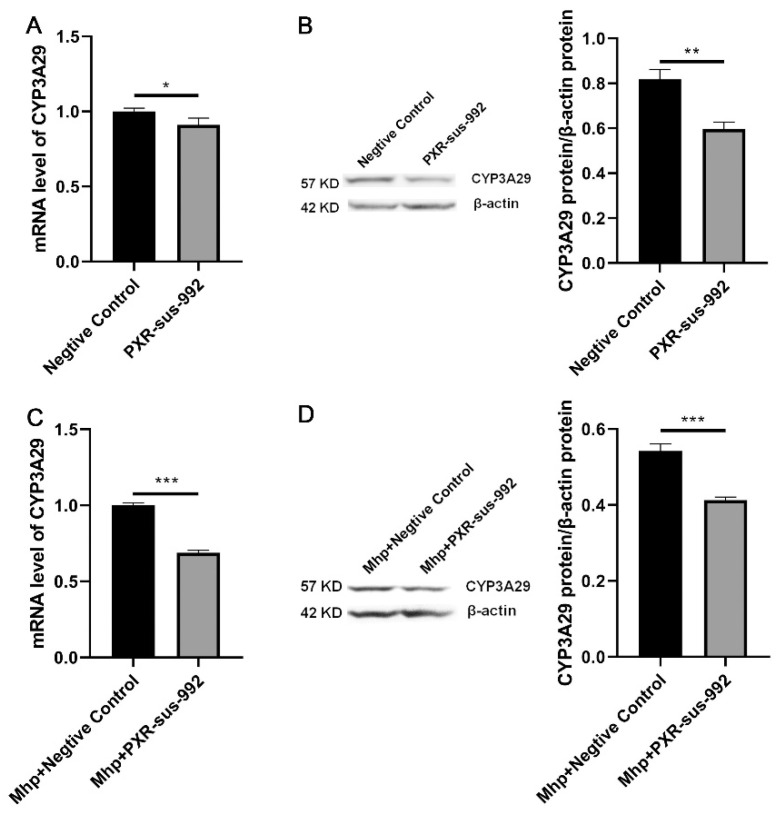
Effect of *PXR* silencing on *CYP3A29* in PAM 3D4/21 cells with or without *M. hyopneumoniae* infection. (**A**) The mRNA expression level of *CYP3A29* after *PXR* silencing. (**B**) Negative control or *PXR*-sus-992 were transfected into PAM 3D4/21 cells. The protein expression of *CYP3A29* was measured by western blot. (**C**) The mRNA expression level of *CYP3A29* after silencing of *PXR* in PAM 3D4/21 cells infected with *M. hyopneumoniae*. (**D**) Negative control or *PXR*-sus-992 was transfected into PAM 3D4/21 cells infected with *M. hyopneumoniae*. The protein expression of *CYP3A29* was measured by western blot. The data are presented as the mean ± SD. The experiment was repeated three independent times. * *p* < 0.05; ** *p* < 0.01; *** *p* < 0.001.

**Figure 6 animals-11-00349-f006:**
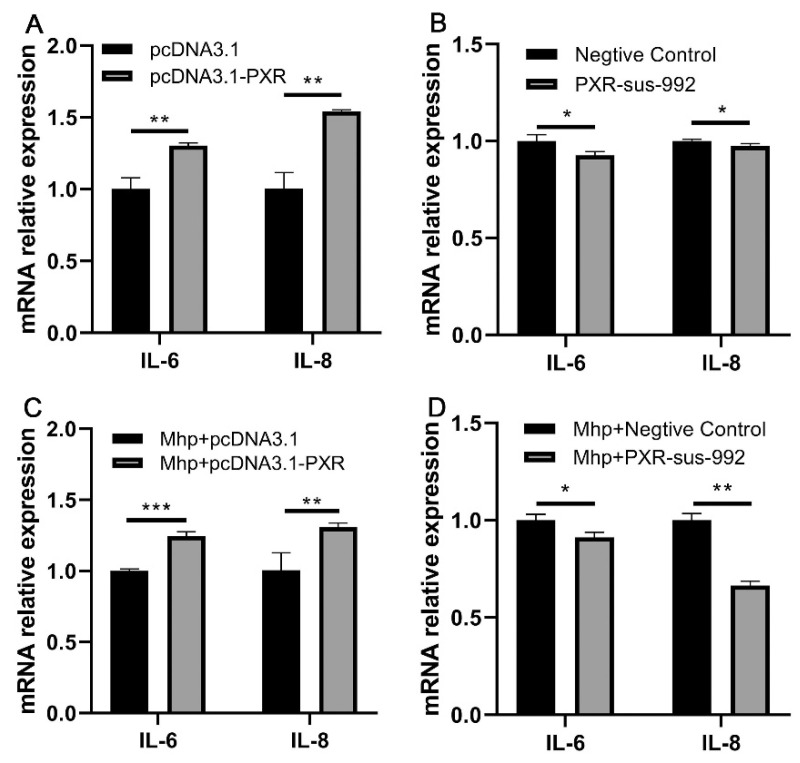
Effects of *PXR* overexpression and silencing on *IL-6* and *IL-8* expression in *M. hyopneumoniae*-infected PAM 3D4/21 cells. (**A**) The mRNA expression levels of *IL-6* and *IL-8* after *PXR* overexpression. (**B**) The mRNA expression levels of *IL-6* and *IL-8* after *PXR* silencing. (**C**) The mRNA expression levels of *IL-6* and *IL-8* after *PXR* overexpression with *M. hyopneumoniae* infection. (**D**) The mRNA expression levels of *IL-6* and *IL-8* after *PXR* silencing with *M. hyopneumoniae* infection. The data are presented as the mean ± SD. The experiment was repeated three independent times. * *p* < 0.05; ** *p* < 0.01; *** *p* < 0.001.

**Figure 7 animals-11-00349-f007:**
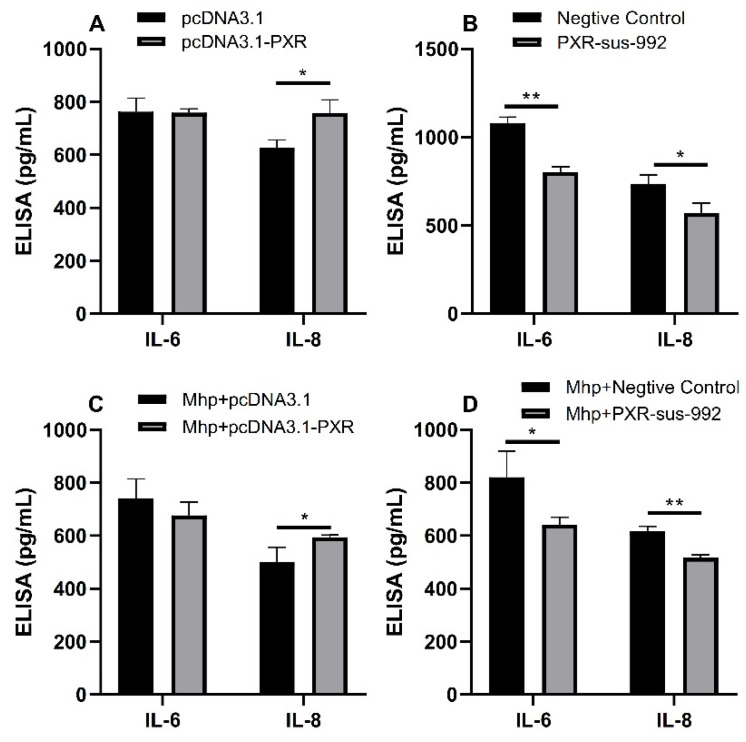
ELISA result of *IL-6* and *IL-8* protein in the cell culture supernatant of the PAM 3D4/21 cells. (**A**) The protein expression of *IL-6* and *IL-8* after *PXR* overexpression. (**B**) The protein expression of *IL-6* and *IL-8* after *PXR* silencing. (**C**) The protein expression of *IL-6* and *IL-8* after *PXR* overexpression with *M. hyopneumoniae* infection. (**D**) The protein expression of *IL-6* and *IL-8* after *PXR* silencing with *M. hyopneumoniae* infection. The data are presented as the mean ± SD. The experiment was repeated three independent times. * *p* < 0.05; ** *p* < 0.01.

**Table 1 animals-11-00349-t001:** Primers used for real-time quantitative polymerase chain reaction (RT-qPCR) amplification.

Primer Name	Primer Sequence (5′→3′)	Length/bp	Annealing Temperature/°C
*IL-6*	CGAGGCCGTGCAGATTAGTA	245	60
	ACGGCATCAATCTCAGGTGC		
*IL-8*	GCAGAGCTCACAAGCTCCTA	171	60
	CTGGCATCGAAGTTCTGCAC		
*CYP3A29*	AAAGTCGCCTCACAGATCAACA	173	60
	GGAGAGAGCACTGCTAGTGGTCT		
*HPRT1*	CCCAGCGTCGTGATTAGTGA	191	60
	TTGAGCACACAGAGGGCTAC		
*PXR*	AGATCTTTTCCCTGCTGCCC	234	60
	CTGAGGGGTCTTCCAAGCTG		
*P36*	TTACAGCGGGAAGACC	427	60
	CGGCGAGAAACTGGATA		

## Data Availability

Data supporting the results of this study shall, upon appropriate request, be available from the corresponding author.

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
