# Peer review of "Effect of Pregnane X Receptor on CYP3A29 Expression in Porcine Alveolar Macrophages during Mycoplasma hyopneumoniae Infection"

_animals, 2021, doi:10.3390/ani11020349_

Round 1

Reviewer 1 Report

I have no other questions and am satisfied with their response to my queries.

Author Response

Thank you very much, sir! We gratefully thanks for the precious time the reviewer spent making constructive remarks!

Reviewer 2 Report

The authors have addressed all the comments to my satisfaction and significantly improved their manuscript.

I accept the authors response defending novelty of their work and withdraw my previous criticism.

I am happy to recommend the article for publication in Animals 

Author Response

(The authors gave the same response as above.)

Reviewer 3 Report

The revised version looks better now.

Author Response

Thank you very much, sir! We gratefully thanks for the precious time the reviewer spent making constructive remarks!

This manuscript is a resubmission of an earlier submission. The following is a list of the peer review reports and author responses from that submission.

Round 1

Reviewer 1 Report

Delete “human” from line 57 in the introduction. It is not “human CYP3A4” that is regulated in pigs.

Why was mRNA expression only measured at 24 hrs post-infection. Is it possible that expression CYP3A4 and cytokine expression occurred earlier than 24 hours?  I mention this because the overall levels of expression, as reported, ate not that high, less than two-fold increase in expression.

The discussion (line 272) regarding the possible role of PPAR-g signaling is speculative without the referred to unpublished data and should not be included in this paper until that work is also published.

Author Response

要点1:从简介中的第57行删除“人”。猪中所调节的不是“人类CYP3A4”。

回应1:非常感谢您的宝贵建议。实际上,我们要表达的是猪的CYP3A29与人的CYP3A4类似。因此,该句改为:“在猪中调节CYP3A29的机制与在人中调节CYP3A4的机制相似”。(第59和60行)

要点2:为什么仅在感染后24小时才测量mRNA表达。CYP3A4和细胞因子的表达是否可能早于24小时出现?我之所以这样说是因为,据报道,总体表达水平没有那么高,表达增加不到两倍。

回应2:感谢您的严格考虑。我们仅在感染后24小时测量了mRNA表达。因为我们先前的研究在Mhp感染后的五个时间点(0小时,4小时,12小时,24小时,48小时)检测到mRNA表达,发现炎症因子(il-6,il-8等)的表达在猪肺炎支原体感染后24小时,显示出最大的炎症反应[1]。我们也参考张的文章[2]。他们在Mhp感染PAM细胞24小时后检测到IL-1β表达。这些参考文献在手稿中。

[1] Fang,XM,Zhao,WM,Fu,YF等。关键词:猪,支原体肺炎,易感性,分子遗传学基础 农业 科学 中国,2015,48(9):2839-2847。DOI:10.3864 / j.issn.0578-1752.2015.14.015。

[2] Zhang Z,Wei Y,Liu B,等。关键词:猪肺炎支原体感染,Hsp90 / Sec22b通过自噬体载体促进猪肺泡巨噬细胞分泌非成熟IL-1β 分子免疫学,2018,101:130-139。DOI:10.1016 / j.molimm.2018.06.265。

第3点:关于PPAR-g信号的可能作用的讨论(第272行)是推测性的,没有提及未发表的数据,在该工作也发表之前,不应包含在本文中。

回应3:非常感谢您的仔细检查。我们已根据您的建议进行了修改。该句子已被删除。

Reviewer 2 Report

Mycoplasma hyopneumoniae (Mhp) of swine, commonly known as swine asthma, is a chronic respiratory infection in pigs. The mechanisms by which the host immunity response is regulated are not yet known. In the presented article, Yang et al aimed to find out how exactly the molecular background in pigs, infected with Mhp is different from the uninfected pigs.

During their previous previous work, based on the results of the microarray study, the authors selected the CYP3A4 izoenzyme as a candidate gene associated with the response to the infection. In the current study, the authors do confirm, using PCR and the porcine alveolar macrophage cell line that, indeed, expression of CYP3A4 is  upregulated with Mhp infection. The authors suggest that there is another key player, PXR gene, regulating CYP3A4 and the overexpression of PXR reduces the level of CYP3A4, mRna and protein as well. They also note that PXR can regulate the mRNA expression level of IL-6 and IL-8.

The major problem with the study is that it lacks novelty, because the gene (CYP3A4)  was already found in the previous study and is simply confirmed here. It was also already known about PXR regulating  cytochrome P-450 and the found interconnection between the  two is not a new  one. In addition, the previous study (microarray) probably had many differentially expressed genes and one would like to understand why exactly only one of them was selected as a candidate gene associated with infection. What about other  genes that were differentially regulated between conditions? Why not test other valid candidates too?

In the discussion section the authors state that the expression of major inflammatory cytokines in PAM cells is promoted by high expression of CYP3A4 caused by Mhp infection via inhibiting the PPAR-g signaling pathway. The following sentence is unexpected:’ the results of this experiment will be submitted to another paper’.

While lacking in overall novelty, the study could benefit from the inclusion of all of the accumulated evidence, without trying to split the data into two separate submissions. 

However, in its current form the study is not suitable for publication.

Author Response

要点1:这项研究的主要问题是它缺乏新颖性,因为该基因(CYP3A4)已在先前的研究中发现,并在此处简单证实。关于PXR调节细胞色素P-450的信息也已经为人所知,并且发现两者之间的互连并不是一个新事物。 

回应1:

我们非常感谢您宝贵的时间,让审稿人做出建设性的评论。首先,在本研究中,我们研究了猪CYP3A29。尽管猪的CYP3A29和人的CYP3A4是同工酶,但它们并不完全相同。

我们实验证明CYP3A29 mRNA和蛋白表达在肺炎支原体感染PAM细胞期间被上调。我们第一个证明猪肺炎支原体可以诱导PAM细胞中PXR表达的上调。同时,我们第一个证明猪肺炎支原体通过PXR调节PAM细胞中CYP3A29的表达。并且先前的研究被用于猪的肝细胞以证明IFN-α通过PXR上调了CYP3A29的表达。但是猪肺炎支原体是病原体,IFN-α是免疫细胞产生的细胞因子。肝细胞的主要功能是参与生物转化和代谢。肺泡巨噬细胞的功能主要涉及先天免疫。

我们首次显示PXR对猪肺炎支原体感染具有促炎作用。因此,这些使我们与以前的研究有所不同。

关于本文的新颖性:

  1. 我们是第一个证明猪猪肺炎支原体感染过程中PXR表达上调的动物。
  2. 我们是第一个证明猪肺炎支原体通过PXR在PAM细胞中上调CYP3A29表达的方法。
  3. 我们首次发现PXR可以调节PAM细胞中猪肺炎支原体感染期间炎性细胞因子(IL-6,IL-8)的表达。
  4. 我们首次发现PXR在PAM细胞感染猪肺炎支原体期间可能通过PPAR调节IL-6和IL-8。

该研究为猪支原体肺炎的治疗和对这种疾病具有抗性的猪的繁殖提供了理论分子基础。

Point 2: The previous study (microarray) probably had many differentially expressed genes and one would like to understand why exactly only one of them was selected as a candidate gene associated with infection. What about other genes that were differentially regulated between conditions? Why not test other valid candidates too?

Response 2: Thank you for this kind advice! Our previous studies (microarray) did find multiple differentially expressed genes. And our previous studies have shown that cytochrome plays an important role in the inflammatory response caused by mycoplasma infection. At the same time, we found that the expression of CYP3A29 was significantly up-regulated after M. hyopneumoniae infection. Therefore, we conducted related studies on CYP3A29.

要点3:在讨论部分中,作者指出,由Mhp感染引起的CYP3A4的高表达通过抑制PPAR-g信号通路来促进PAM细胞中主要炎症细胞因子的表达。以下句子是意外的:“此实验的结果将提交给另一篇论文”。

回应3:先生,谢谢您的建议!我们已根据您的意见修改了该表达式。该句子已被删除。

Reviewer 3 Report

Manuscript: Effect of pregnane X receptor on CYP3A29 expression in porcine alveolar macrophages during Mycoplasma hyopneumoniae infection In the current manuscript, the authors tried to investigate the mechanisms underlying the upregulation of CYP3A29 in porcine alveolar macrophages after Mycoplasma hyopneumoniae infection. They found that PXR may regulate the expression of CYP3A29 during M. hyopneumoniae infection in macrophages. This work provided a potential mechanism for swine with M. hyopneumoniae infection. Some concerns should be addressed before publication. Major concerns: 1. In the results section 3.1, the authors should introduce the background of the P36 gene and the correlation with M. hyopneumoniae infection. 2. In the results section 3.2, authors should also check the mRNA and protein levels of both PXR and CYP3A29 increased after M. hyopneumoniae infection at 48h and 72h to further support your conclusion. 3. In the results section 3.3.1, Figure 3 labels do not match with the descriptions in the text, such as Figure 3.BC and Figure 3.DE. 4. In Figure 4, authors should combine the results with and without M. hyopneumoniae infection to show CYP3A29 further increase with M. hyopneumoniae infection. And To do the same analysis for Figure 5. 5. To check the IL-6 and IL-8 protein levels by ELISA in Figure 6. Minor concerns: 1. To be careful to use the "significantly" word in the manuscript, such as in Figure 2C, the result doesn't show PXR significantly increase.

Author Response

要点1:在结果部分3.1中,作者应介绍P36基因的背景以及与猪肺炎支原体感染的相关性。

回应1:我们介绍了P36基因的背景以及与猪肺炎支原体感染的相关性。增加了一个新的句子:“ P36是猪肺炎支原体基因组编码的一种免疫显性蛋白,并且已显示p36基因是猪肺炎支原体的特异性。”(第148-150行)

要点2:在结果第3.2节中,作者还应检查猪肺炎支原体感染后48h和72h PXR和CYP3A29 mRNA和蛋白水平的升高,以进一步支持您的结论。

回应2:非常感谢您的宝贵建议。实际上,我们先前的研究在感染Mhp 48小时后检测到相关基因(IL-6,IL-8等)的mRNA表达。结果表明,与24 h相比,在48 h炎症细胞因子的表达被下调。它表明,在24小时时,细胞具有最大的炎症反应。长期培养会导致高细胞密度并对细胞生长产生不利影响。因此,我们没有补充实验来检查猪肺炎支原体感染后48h和72h PXR和CYP3A29的mRNA和蛋白水平升高。

第3点:在结果部分3.3.1中,图3标签与文本中的描述不匹配,例如图3.BC和图3.DE。

回应3:感谢您的提醒!此错误已在修订的稿件(第186和188行)中得到纠正。

要点4:在图4中,作者应结合有和没有猪肺炎支原体感染的结果,显示CYP3A29随着猪肺炎支原体感染的进一步增加。并对图5做同样的分析。

回应4:谢谢您的好建议。我们已经在修订稿中进行了修订。“当将图4. AB与图4. CD进行比较时,发现猪肺炎支原体对CYP3A29的PXR过量表达的促进作用低于猪肺炎支原体感染的对CYP3A29的促进作用”(第203-205行)。并比较了图5. AB和图5. CD,发现肺炎支原体对PXR沉默对CYP3A29的抑制作用比没有肺炎支原体感染的要高。它表明CYP3A29在猪肺炎支原体感染后进一步增加。”(第209-211行)。

第5点:通过图6中的ELISA检查IL-6和IL-8蛋白水平。

回应5:非常感谢您,先生!根据您在图7中的建议,我们补充了ELISA测试以检查IL-6和IL-8蛋白水平(第248-263行)。我们还添加了部分讨论(291-297行)

要点6:要小心使用稿件中的“显着”字词(例如,图2C),结果未显示PXR显着增加。

回应6:先生,谢谢您的建议!经修订的手稿中的“显着”已删除。(第169、236、239行)
